# Field and Numerical Study of Resistance and Resilience on a Sea Breeze Dominated Beach in Yucatan (Mexico)

**Gabriela Medellín** [1,2,*] **, Alec Torres-Freyermuth** [1,2] **, Giuseppe Roberto Tomasicchio** [3] **, Antonio Francone** [4] **, Peter A. Tereszkiewicz** [5] **, Letizia Lusito** [3] **, Leonardo Palemón-Arcos** [6] **and José López** [1,2]

[1] Laboratorio de Ingeniería y Procesos Costeros, Instituto de Ingeniería, Universidad Nacional Autónoma de México, Sisal, Yucatán 97835, Mexico; atorresf@iingen.unam.mx (A.T.-F.); jlopezgo@iingen.unam.mx (J.L.)
[2] Laboratorio Nacional de Resiliencia Costera, Laboratorios Nacionales CONACYT, Sisal, Yucatán 97835, Mexico
[3] Department of Engineering, University of Salento, 73100 Lecce, Italy; roberto.tomasicchio@unisalento.it (G.R.T.); letizia.lusito@unisalento.it (L.L.)
[4] Department of Engineering, University of Calabria, 87036 Arcavacata di Rende, Italy; antonio.francone@unical.it
[5] Department of Earth and Environmental Sciences, University of West Florida, Pensacola, FL 32514, USA; petert@email.sc.edu
[6] Departamento de Ingeniería Civil, Universidad Autónoma del Carmen, Cd. del Carmen, Campeche 24180, Mexico; leopalemon@hotmail.com
* Correspondence: gmedellinm@iingen.unam.mx; Tel.: +52-988-931-1000

**Abstract:** The understanding of the beach capability to resist and recover from a disturbance is of paramount importance in coastal engineering. However, few efforts have been devoted to quantifying beach resilience. The present work aims to investigate the shoreline resistance and resilience, associated to a transient disturbance, on a sandy beach. A temporary groin was deployed for 24 h on a micro-tidal sea-breeze dominated beach to induce a shoreline perturbation. Morphological changes were measured by means of beach surveys to estimate the beach perturbation and the further beach recovery after structure removal. An Empirical Orthogonal Function (EOF) analysis of the shoreline position suggests that the first EOF mode describes the spatial-temporal evolution of the shoreline owing to the groin deployment/removal. A new one-line numerical model of beach evolution is calibrated with the field surveys, reproducing both the sediment impoundment and subsequent beach recovery after the structure removal. Thus, a parametric numerical study is conducted to quantify resistance and resilience. Numerical results suggest that beach resistance associated to the presence of a structure decreases with increasing alongshore sediment transport potential, whereas resilience after structure removal is positively correlated with the alongshore diffusivity.

**Keywords:** beach resilience; beach resistance; temporary groin; sea breezes; resilience index; GSb model; Yucatan peninsula

## 1. Introduction

The stability of an ecosystem depends on both resistance and resilience capability to withstand a given perturbation associated to either natural or anthropogenic disturbances [1–3]. The resilience concept has been widely employed in ecological [4] and social [5] sciences and disaster risk reduction [6,7]. However, studies incorporating resilience for coastal engineering applications are scarce [8–10] and hence further research is needed [11].

The beach resistance in the coastal vulnerability context can be associated to the amount of change produced by wave events and/or due to the presence of coastal infrastructure which alter the mean pattern of sediment transport in the coastal environment; thus, it is a measure of the beach capability to resist deviations with respect to an equilibrium morphological condition. On the contrary, the beach resilience determines the speed with which the beach morphology features (e.g., dune elevation and shoreline position) return to the pre-disturbed condition [2]. Thus, the knowledge of the beach stability (i.e., resistance and resilience) is fundamental for decision-making regarding mitigation measures against beach erosion. Beach erosion in the northern Yucatan coast is critical at many locations owing to the presence of coastal structures [12,13]. Meyer-Arendt [14] reported that construction of traditional groins, made of timber and rocks, began in the 1950s and increased significantly in the late 1960s with the construction of the port infrastructure. Furthermore, beach erosion has been exacerbated during the past decade owing to the use of impermeable groins and breakwaters. Therefore, structure removal has been considered as a mitigation measure against beach erosion in this region [13]. However, no information regarding the mechanisms controlling the recovery of the shoreline position after structure removal is available.

Temporary groins have been applied in previous studies [15–17] to measure alongshore sediment transport and to calibrate sediment transport formulations. More specifically, the beach morphology changes measured in such studies have been considered for the estimation of the $K$ parameter in the CERC equation [18]. However, less efforts have been devoted to investigating beach evolution after coastal structures removal. Recent studies have focused on investigating the morphodynamic responses to seafloor artificial perturbations (e.g., excavated holes and channels) in the nearshore [19–21]. Moulton et al. [20] investigated the mechanisms controlling the infill of large excavated holes in the surf zone, finding that downslope gravity-driven bedload transport was important in morphological evolution for bathymetric features with large slopes.

The present work aims to investigate the shoreline resistance and resilience on a sea-breeze dominated beach by means of field observations and numerical modelling. The main findings are that, on sea breeze dominated environments, the: (i) beach resistance to the presence of a groin is negatively correlated with the alongshore sediment transport potential; and (ii) beach resilience after the structure removal is positively correlated with alongshore diffusivity.

The outline of the paper is as follows. Firstly, the study area is presented in Section 2. The experimental setup, numerical model and data analysis are described in the Materials and Methods section (Section 3). Section 4 presents the field observations during the experiment and the numerical model calibration and verification. Then, a discussion on the mechanisms controlling the shoreline resistance and resilience is presented (Section 5). Finally, concluding remarks are given in Section 6.

## 2. Study Area

The study area is located on a barrier island in the northern Yucatan Peninsula (Figure 1a), at the fishing village of Sisal (see Figure 1b). This coastal region is characterized by a micro-tidal range, intense sea breeze conditions, a mild continental shelf and low energy waves [22]. The field experiment was conducted between the Port and the Sisal Pier (see Figure 1c). Winds, offshore waves and mean sea level have been measured over the past years in order to characterize the main forcing mechanisms affecting the coastal region in this area (Figure 1b,c).

Wind conditions in the study area are dominated by synoptic scale patterns (i.e., Bermuda-Azores, easterly winds, cold-fronts and tropical waves) and local sea breezes [23]. The NE sea breeze winds are present throughout the year but are more frequent and intense in May. On the other hand, Central America Cold Surge events, associated with cold-front passages, are more frequent during winter months. Cold-fronts, usually originated in the Rocky Mountains [24], are characterized by sustained winds ($W > 15$ m s$^{-1}$) from the NNW and a high-pressure system (Figure 2a). Therefore, the mean wave climate in the study area is associated to locally generated NE waves owing to sea breeze events, with significant wave height $H_s < 1$ m (at $h = 10$ m water depth) (Figure 2b). More energetic NNW

swell waves ($H_s$ > 2 m and $T_p$ > 7 s), associated to cold-fronts, occur during winter months (see Figure 2b). Furthermore, the presence of tropical storms is ubiquitous in the area [25,26]. Tidal regime is mixed, predominantly diurnal, with a spring and neap tidal range of 0.8 m and 0.1 m, respectively [27].

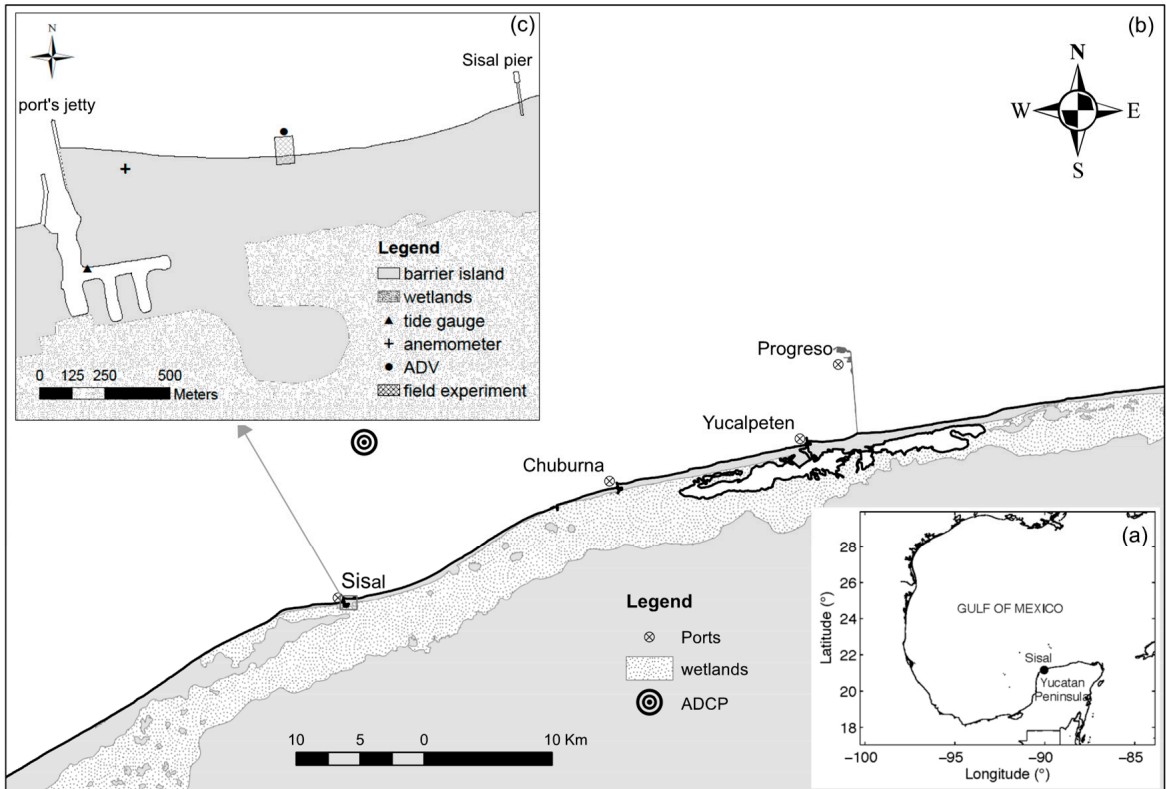

**Figure 1.** Location map showing (**a**) the Yucatan peninsula at the SE of the Gulf of Mexico, (**b**) a section of the Yucatan north coast showing the barrier island and wetlands and (**c**) the study area location, existing monitoring systems (ADCP: Acoustic Doppler Velocimeter Profile; ADV: Acoustic Doppler Velocimeter) and coastal structures.

Torres-Freyermuth et al. [22] conducted a field experiment to characterize the nearshore circulation during both intense local sea breeze events and synoptic Norte events. They found that during intense sea breeze events the alongshore currents significantly increase inside the surf and swash zones. Therefore, in the present case of a coast subjected to sea breeze conditions, the highly oblique winds (see Figure 2a) play a major role in driving longshore currents and consequently longshore transport [28–30].

The beach in the study area is composed of sand with median grain size, $D_{50}$, equal to 0.3 mm [31]. Furthermore, it presents a shoreline orientation 14° south of the E-W orientation [22], which is altered near the port's jetty and the Sisal Pier. The nearshore bathymetry is characterized by the presence of a sand bar system (Figure 3), where the outer bar is relatively alongshore uniform and inner bars present a high seasonal variability. Analysis of the shoreline variability, during the intense sea breeze season (May to September, 2015), at two transects bounding the study area suggests a small (<3 m) cross-shore variation at these locations (not shown). Therefore, the temporal groin experiment was conducted between these two transects located at the middle section between the two structures (Figure 3).

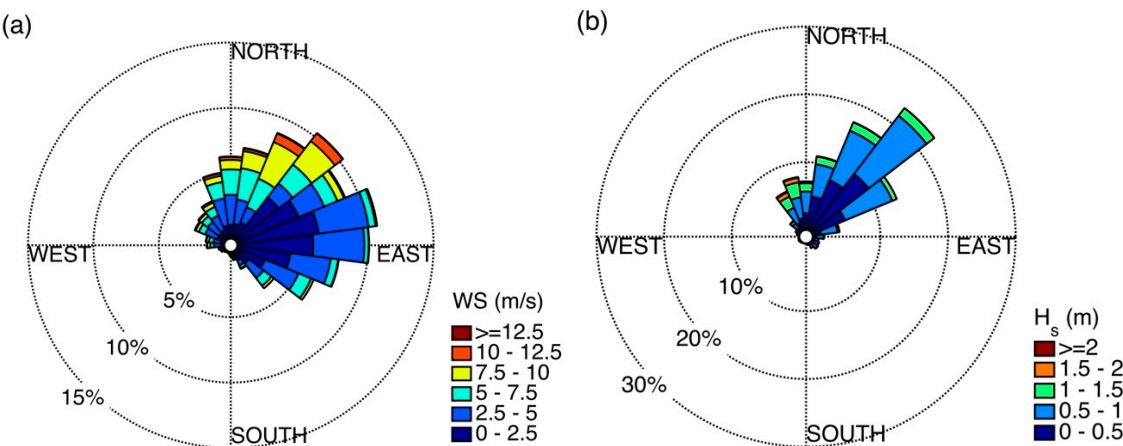

**Figure 2.** (**a**) Wind and (**b**) wave roses in the study area. Wind data are taken from the Sisal weather station MeteoSisal (www.weatherunderground.com 1 January 2009–17 June 2016) and the wave data were collected from an ADCP located at 10 m water depth in front of Sisal (10 December 2013–20 April 2016). *WS* and $H_s$ stand for wind speed and significant wave height, respectively.

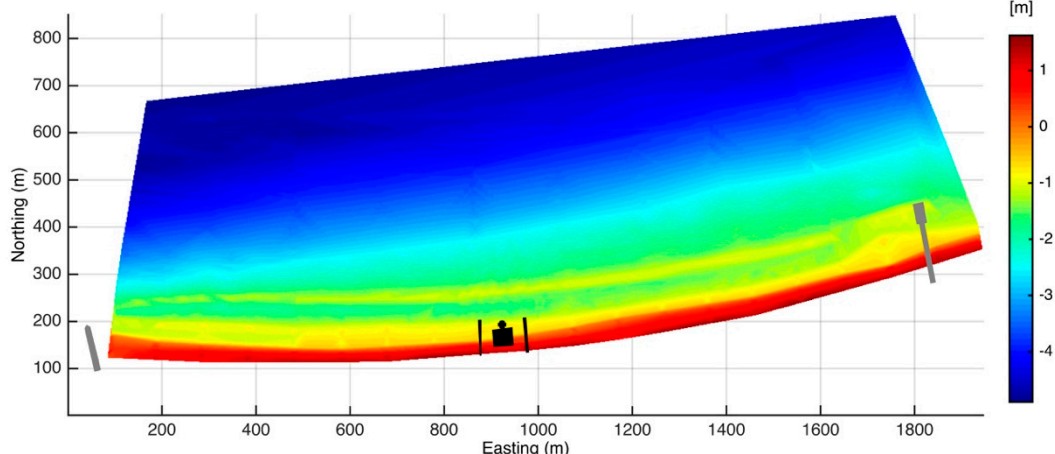

**Figure 3.** Bathymetry measured two days before the experiment (25 May 2015), showing the study area (■), the ADV (●) and the updrift/downdrift control lines (black solid lines) locations. The study site is located between the jetty of the Sisal port entrance channel (left-hand side, gray line) and the Sisal pier (right-hand side gray feature). The color bar indicates the elevation with respect to the mean sea level.

## 3. Materials and Methods

A description of field observations and data analysis is presented in this section. Furthermore, the numerical model employed in this work is also described.

### 3.1. Field Experiment

The field experiment was conducted in Spring 2015 to investigate the beach stability owing to the presence/removal of an artificial perturbation in the swash zone. Beach surveys before, during and after the structure deployment allow us to investigate shoreline resistance and resilience. We focused on a short-term sea breeze event due to: (i) the important role that sea-breeze events play in the sediment transport in the study area; (ii) the difficulties for conducting the beach surveys during more energetic wave conditions (storm conditions); and (iii) the labor-intensiveness required for obtaining high- spatial and temporal resolution morphology data for a longer period.

### 3.1.1. Temporary Groin

The temporary groin built for this study was based on the design proposed by [16]. The groin dimensions are consistent with the typical structures found along the northern Yucatan coast. The structure was made of 0.19 m thick wood-sections of dimensions 2.40 m by 1.20 m, with holes (0.08 m diameter) drilled and covered with a 63 µm sieve cloth in order to avoid a returning offshore flow near the structure [16]. The groin consisted of seven wood sections, installed over a frame made of iron pipes and clamps; 10 m are inside the surf/swash zone within the region of active transport and the remaining 4.4 m lie on the dry beach resulting on a total length of 14.4 m. Furthermore, sandbags were spread out along the base of the structure to avoid bed scouring that could lead to sediment bypassing underneath the structure. The sand bags consisted of polypropylene woven raffia bags (60 by 100 cm) filled at approximately 2/3 of their capacity with sand.

The deployment of each section started from the land toward the sea, allowing a 0.20 m overlap between sections. Each section consisted of three vertical pipes pounded 1.5 m into the sand bed using a hammer and one horizontal pipe, holding the three pipes with scaffold clamps (Figure 4a). The original design from [16] was improved by including two horizontal members, at the down-drift side of the structure (see Figure 4a), perpendicular to the groin and attached with clamps to an additional scaffold frame. This design provided additional resistance to alongshore forces induced by wind, waves and currents. The groin deployment took approximately three hours for a team of 12 people.

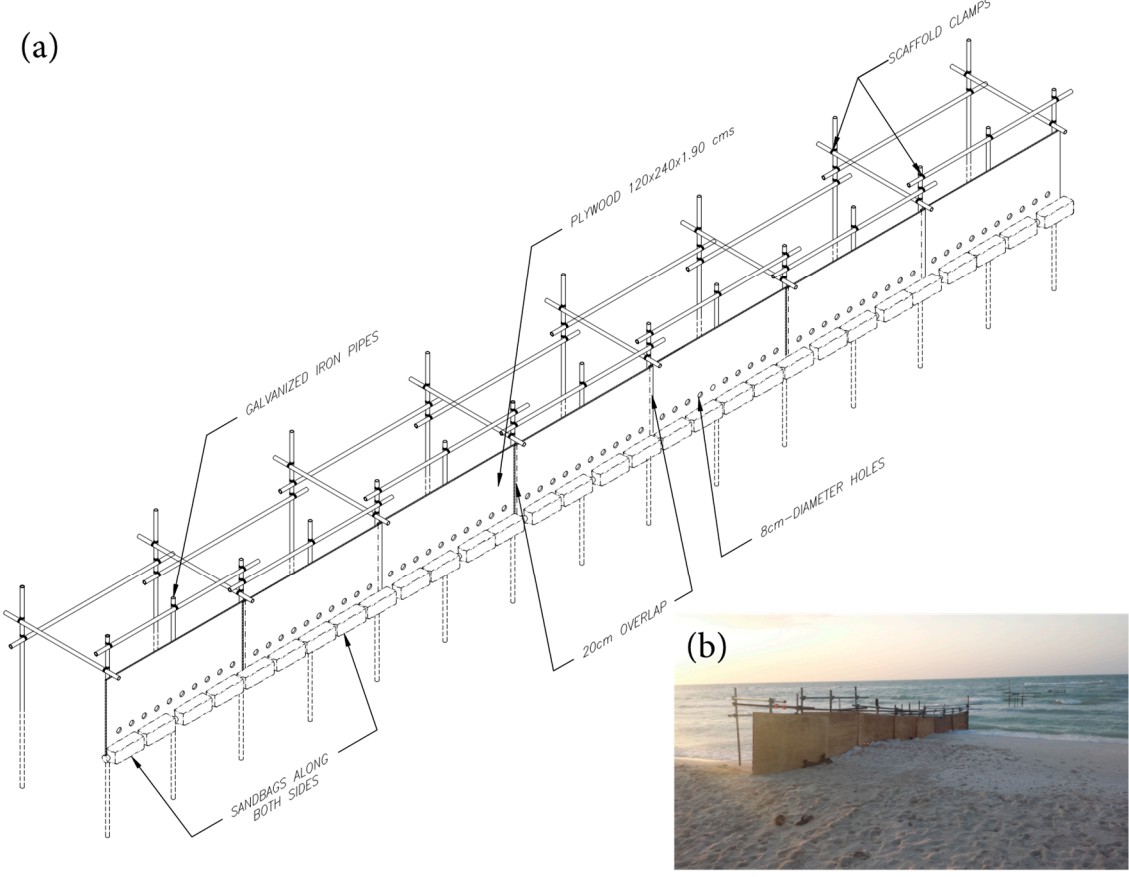

**Figure 4.** (**a**) Temporary groin design made of wood-sheets, lined with sand bags and a scaffold frame of iron pipes. (**b**) Picture of the shoreline perturbation 12-h after the groin deployment.

### 3.1.2. Data Collection

Different sensors were deployed to characterize the environmental conditions occurring during the field experiment. Wind data was measured every minute using a weather station located in a

tower installed near the Sisal Port (Figure 1c). Offshore wave conditions were recorded at 10 m water depth using an RDI Acoustic Doppler Current Profiler (ADCP) located 11 km offshore (see Figure 1b for instrument location). During the experiment, three breaker lines were observed at the outer and inner bars and the inner-surf/swash zone transition. Moreover, an Acoustic Doppler Velocimeter (ADV) Nortek Vector, located onshore the inner bar (Figure 5a,b) at 0.2 m above the seabed, acquired high-frequency (16 Hz) velocity measurements during 48 h (27 May to 29 May). The instantaneous velocities were measured in the *XYZ* coordinate system, where velocities are defined such that *u*, *v* and *w* velocities correspond to the *x* (cross-shore), *y* (alongshore) and *z* (vertical) directions, respectively.

A temporary groin was deployed in the inner surf/swash zone on the morning (0800 local time) of 27 May and was removed 24-h later on the morning of 28 May 2015. Beach morphology was surveyed along 15 survey lines, covering the up- and down- drift sides of the temporary structure (Figure 5a). A Leica Differential Global Positioning System (DGPS) was employed using Real Time Kinematics (RTK) for conducting high-resolution topographic surveys. The equipment in RTK mode and Kinematic (phase) moving mode has a horizontal and vertical accuracy of 10 mm and 20 mm, respectively. The alongshore distance between transects varies from 2 to 6 m, with the highest resolution corresponding to those transects located close to the structure (Figure 5a). The DGPS beach surveys were conducted every two hours for 24-h to evaluate the beach resistance owing to the structure presence. Furthermore, measurements continued after the structure removal, with the same two-hour temporal resolution for 10 h and then were resumed with a lower temporal resolution (i.e., 29 May, 3 June), continuing until the beach was fully recovered. A total of 20 beach surveys were conducted. Control lines, located 50 m updrift and downdrift from the structure location (Figure 5a), were surveyed weekly to assess the natural beach variability in this area. It is worth to notice that the survey lines only cover the swash and inner surf zone and hence do not cover the entire surf zone which extends offshore. The field data is available via author's request following the instruction in the supplementary material section.

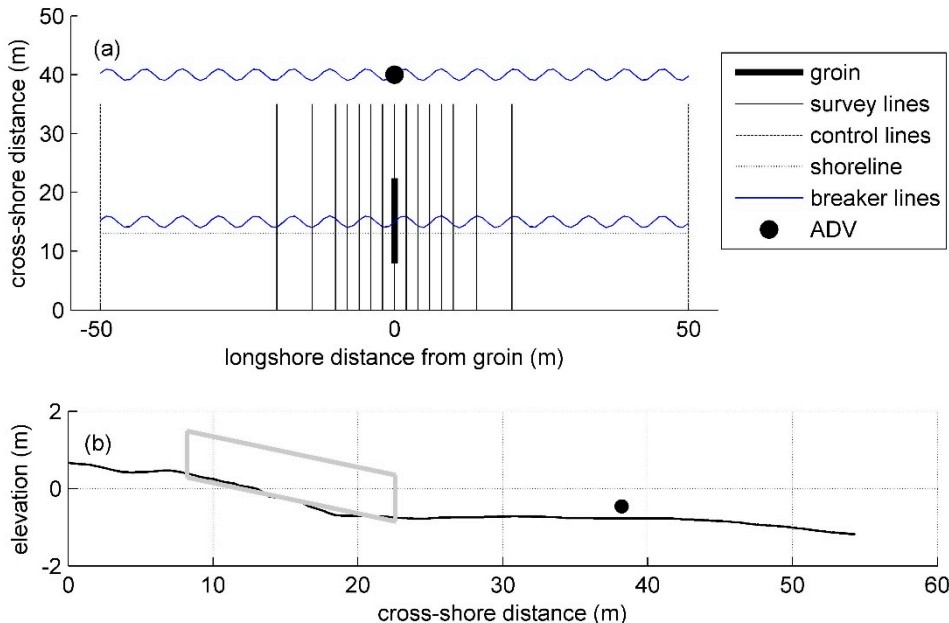

**Figure 5.** (**a**) Plan view of the survey lines and the structure location, showing the breaker lines and ADV location. (**b**) Beach profile of the middle transect showing the ADV location with respect to the structure.

The mean sea level was acquired every minute by a tidal gauge located inside the Sisal Port (Figure 1c). Mean sea level in the study area presents a cyclic annual variation [32], showing a minimum in July and a maximum in October. Therefore, when a short-term experiment is carried out (days to

weeks), the seasonality of the mean sea level (msl) elevation should be considered. The vertical datum of the surveys corresponds to the MEX97 geoid [33] and the difference between the vertical datum and the mean water elevation during the experiment is approximately 0.2 m. Beach surveys in the present paper are referenced to the msl during the experiment period (May–June 2015). A summary of the field data collected is presented in Table 1.

**Table 1.** Measured data, sensor employed, sampling frequency and measured period.

| Variable | Sensor | Manufacturer | Sampling Frequency | Start Date | End Date |
|---|---|---|---|---|---|
| Morphology | GPS | Leica Geosystems (Gallen, Switzerland) | 2–24 h | 27 May 2015 | 3 June 2015 |
| Waves | ADCP | RD Instruments (Poway, CA, USA) | 30 min | 27 May 2015 | 3 June 2015 |
| Surf zone currents | ADV Vector | Nortek (Rud, Norway) | 16 Hz | 27 May 2015 | 29 May 2015 |
| Sea level | Radar level sensor | OTT HydroMet (Kempten Germany) | 1 min | 27 May 2015 | 4 June 2015 |
| Winds | Weather station | Davis Vantage Pro 2 (Hayward, CA, USA) | 1 min | 27 May 2015 | 4 June 2015 |

### 3.1.3. Field Data Analysis

The shoreline position ($z = 0$) was extracted from each of the cross-shore DGPS survey lines. An Empirical Orthogonal Function (EOF) analysis of the shoreline position [34–36], with respect to the initial shoreline, was performed in order to investigate the dominant modes of variability during the experiment. The EOF analysis allows to represent the coastline data, $x(y, t)$, as a summation of $n$ spatial ($e_n(y)$) and temporal functions ($c_n(t)$), where the variance explained decreases with the mode number, as follows:

$$x(y, t) = \sum_{n=1}^{N} c_n(t) e_n(y)$$

The shoreline variability associated to the presence of the temporal groin can be evaluated by analyzing the resulting spatial and temporal functions.

Wind magnitude and direction, from the meteorological tower, were averaged every 5 min. Wave statistics measured by the ADCP were computed every 60 min based on pressure and velocity measured at 2 Hz. On the other hand, the ADVs velocity information was removed when the correlation value was less than 80% with the aim of identifying potentially inaccurate measurements. The measured velocities were averaged over 512-s intervals to ensure stationary conditions and further estimate the (cross- and along- shore) currents inside the surf zone.

### 3.2. Numerical Model

### 3.2.1. Model Description

Numerical simulations have been conducted by means of a newly proposed morphodynamic model, named General Shoreline beach 1.0 (GSb), belonging to the one-line model typology [37]. This typology assumes that the beach cross-shore profile remains unchanged [38,39], thereby allowing beach change to be described uniquely in terms of the shoreline position. The peculiarity of the GSb model consists of simulating shoreline evolution based on a longshore transport formula/procedure suitable at any coastal mound: sand, gravel, cobbles, shingle and rock beaches [40–44]. The GSb model presents one calibration coefficient solely, $K_{GSb}$, which does not depend on the grain size diameter and depends on the alongshore gradient in breaking wave height [45]. The proposed general formula/procedure considers an energy flux approach combined with an empirical/statistical relationship between the wave-induced forcing and the number of moving units. GSb model allows to determine short-term (daily base) or long-term (years base) shoreline change for arbitrary combinations and configurations of structures (groins, jetties, detached breakwaters and seawalls) and beach fills that can be represented on a modelled reach of coast. A demo version of the numerical model can be downloaded by following the instructions in the supplementary material section.

### 3.2.2. Data Analysis

The results from the numerical simulations have been adopted to investigate the shoreline beach resistance and resilience at the considered stretch of coast. The resistance index, $RS$ and resilience index, $RL$, proposed by [2], to investigate ecological stability, were adapted for the present study. $RS$ is then defined as,

$$RS(t_0) = 1 - \frac{2|\Delta S_0|}{(l + |\Delta S_0|)} \tag{1}$$

where $\Delta S_0$ represents the cross-shore distance, at $t_0 = 720$ h, between the perturbed shoreline and the unperturbed shoreline, in vicinity of the groin and $l$ is the perturbation length which equals to the groin length. $RS$ ranges between 0 and 1; minimal resistance (largest effect) to the beach perturbation corresponds to smaller values of $RS$. Similarly, the resilience index $RL(t)$ is defined as,

$$RL(t) = \frac{2|\Delta S_0|}{(|\Delta S| + |\Delta S_0|)} - 1 \tag{2}$$

where $\Delta S$ represents the cross-shore distance, at time $t$, between the perturbed shoreline and the unperturbed shoreline with $RL = 1$ corresponding to a fully recovered shoreline (i.e., return to the pre-disturbed condition). Shoreline beach resistance and resilience are illustrated in Figure 6.

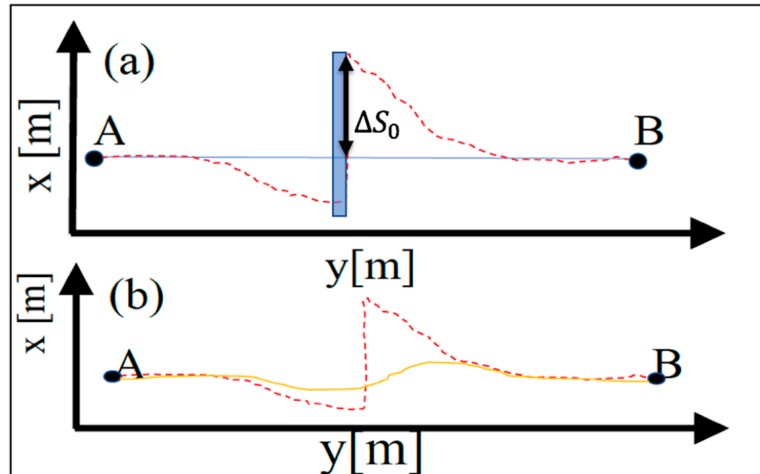

**Figure 6.** Definition sketch of shoreline evolution between points A and B for the cases of: (**a**) beach resistance owing to the groin (original shoreline: blue line; disturbed shoreline: red dashed line) and (**b**) beach resilience after the structure removal (perturbed shoreline: red-dashed line; shoreline position at time $t_i$ after structure removal: yellow line).

## 4. Results

### 4.1. Field Observations

#### 4.1.1. Forcing Conditions

Wind velocity time series shows a diurnal variability associated to sea-breeze events (Figure 7a). A maximum wind speed of 16 m s$^{-1}$, corresponding to the sea breeze event peak, was recorded during the experiment in the afternoon (1630 local time) of 27 May (Figure 7a). Offshore wave conditions, measured at 10 m water depth (Figure 7b), are highly correlated with local winds (Figure 7a). The wave height increased from $H_s = 0.3$ m measured at 10:00 to $H_s = 1.0$ m measured at 1800, with mean wave direction approaching from the NE (Figure 7b). The temporary structure was deployed during neap tides (light gray shade in Figure 7c) in order to decrease the influence of the tide on the effective length of the groin and hence restricting the swash zone width [16].

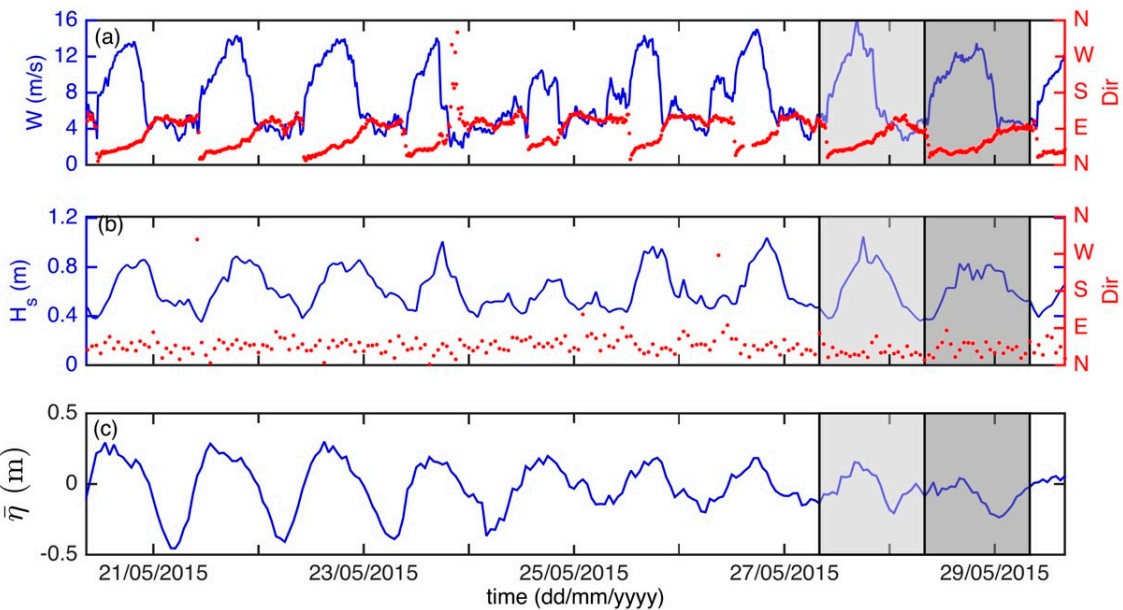

**Figure 7.** Measured (**a**) wind speed and wind direction (**b**) offshore significant wave height and wave direction at 10 m water depth and (**c**) mean sea level variation $\bar{\eta}$ from the Sisal gauge before, during (light gray shade) and after (dark gray shade) the groin deployment.

Sea breeze events increased the wave energy inside the surf zone (Figure 8a). The alongshore current velocity $V_y$ becomes negligible during the land breeze (0300 h), whereas it reaches $V_y > 0.3$ m s$^{-1}$ near the sea breeze peak (1800 h) (Figure 8b). Thus, westward currents dominated the surf zone hydrodynamics during the measured period (red solid-line in Figure 8b). On the other hand, the mean cross-shore current $V_x$ (blue solid-line in Figure 8b) shows negligible current intensity at this location/elevation during the 48 h period.

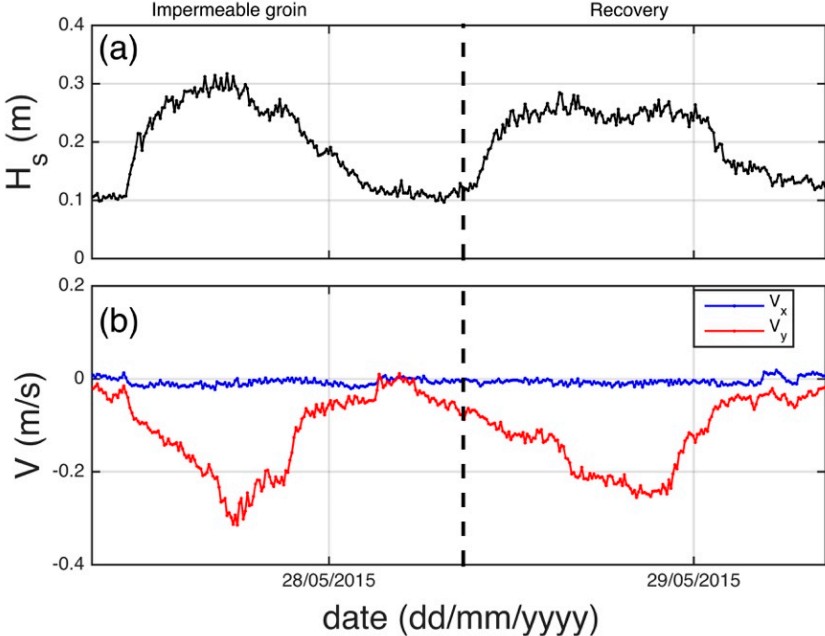

**Figure 8.** Surf zone conditions during and 24 h after the structure disturbance. (**a**) Significant wave height inside the surf zone and (**b**) cross- (blue line) and along- (red line) shore currents. Negative values in panel (**b**) indicate offshore/west-ward currents, while positive values indicate onshore/east-ward currents.

4.1.2. Observed Morphology Evolution

Beach morphology changes are evaluated by analyzing the high spatial and temporal resolution RTK DGPS survey data (Table 1 and Figure 9). The measured beach survey before the structure deployment (0730 local time) shows the alongshore uniformity of the beach contour lines (Figure 9a). However, the deployment of a temporary impermeable structure (at $x = 0$ m) induces a shoreline perturbation owing to the sediment impoundment at the east side of the structure and erosion at the downdrift (west) side (a time-lapse video of the experiment is included as supplementary material). The beach survey conducted 24 h after the structure deployment (0842 of the 28 May 2015) shows significant changes for $h > -0.6$ m (Figure 9b). These observations are consistent with the westward alongshore current associated to sea breeze events (Figure 8b). The shoreline contour reaches a maximum advance of 6 m and maximum retreat of less than $-4$ m at the updrift and downdrift side, respectively. The calculated sediment volume impoundment, at the updrift side of the structure, reached 70 m$^3$ in 24 h (2 to 6 m$^3$/m), whereas the volume loss at the downdrift for the same period is less than 40 m$^3$. Differences between up- and down- drift volume changes might be ascribed to the limited alongshore spatial coverage of the topographic measurements.

Alongshore uniformity for the submerged area ($h < 0$ m) was observed 24 h after the structure removal (0825 of the 29 May 2015). However, a clear perturbation in the subaerial beach profile was still present (Figure 9c). The disturbance smooths out during the following days, returning to the pre-disturbed condition (i.e., straight and parallel contours) 144 h after the structure removal (3 June 2015) (Figure 9d).

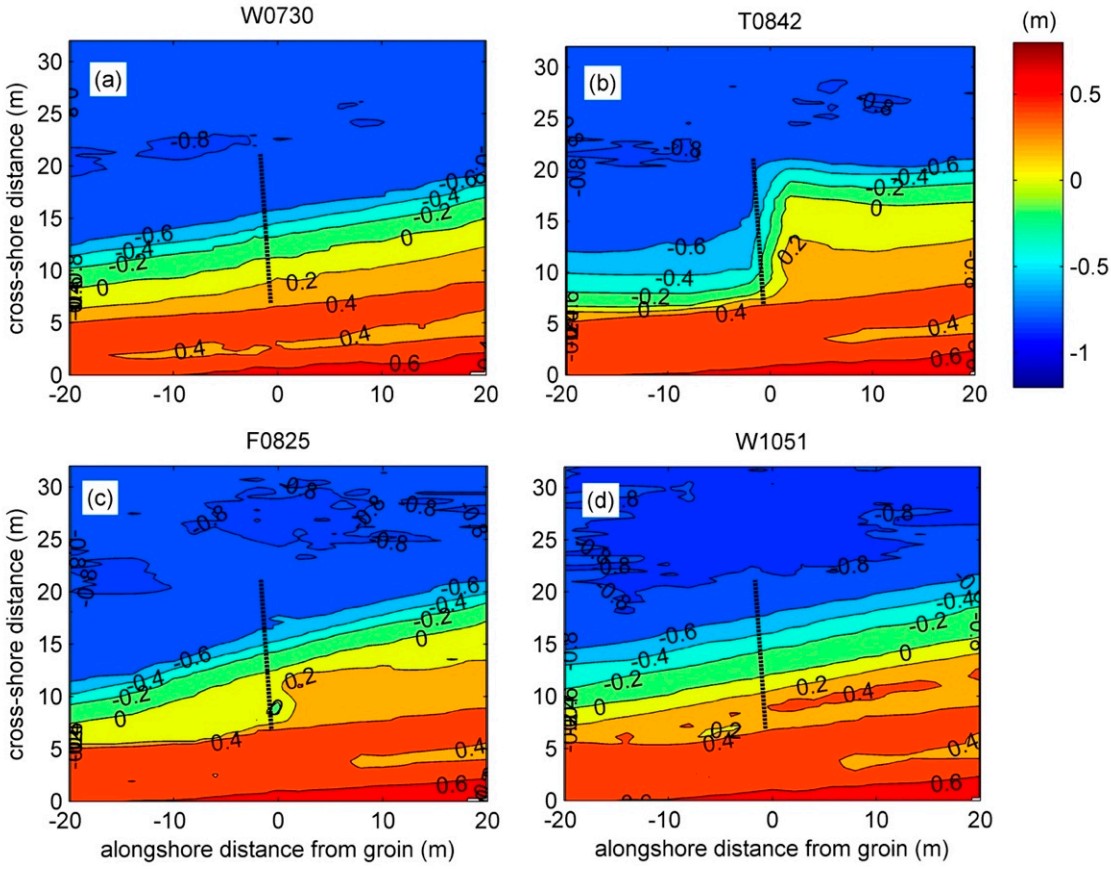

**Figure 9.** Beach survey (**a**) right before groin deployment, (**b**) right before groin removal (24 h later), (**c**) 24 h after groin removal, (**d**) 144 h after groin removal. The title of each subplot indicates the weekday and local time of each survey (W = Wednesday, T = Thursday, F = Friday).

The EOF analysis of the shoreline position allowed us to identify the spatial and temporal shoreline evolution before and after the structure removal. The first mode of variability, which represents more than 91% of the total variance, has been associated solely to the groin influence. The spatial function of the first mode ($e_1(y)$ in Figure 10a) describes the shoreline perturbation with positive/negative values associated to accretion/erosion at the updrift/downdrift side of the temporal groin. Furthermore, the temporal evolution ($c_1(t)$ in Figure 10b) shows an increase from zero to a maximum value in the first 24 h (before structure removal). After the structure removal, $c_1(t)$ shows slower decrease reaching a value close to zero at the end of the measured period. The second and third modes (not shown) represent 7.5% of the total variance and show spatial and temporal functions not associated to the groin presence. The beach recovery time can be calculated from the decay time of the perturbation described by the first temporal function ($c_1(t)$) which is nearly 144 h.

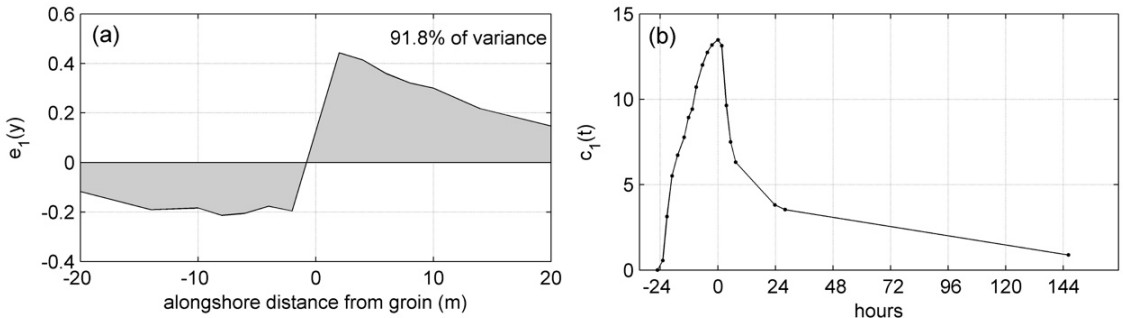

**Figure 10.** Empirical orthogonal function analysis of the shoreline data during the experiment (27 May–3 June, 2015). Only the (**a**) spatial function and (**b**) the temporal evolution of the first mode are shown.

## 4.2. Numerical Modelling

The field data were employed to calibrate the GSb model. Thus, a 14.4 m long groin (10 m wet plus 4.4 m dry) was positioned at the center of the domain. The alongshore model computational domain was assumed equal to 41 m. Model grid cell resolution, *DX*, has been set equal to 1 m with a total number of cells, *NX*, equal to 41, whereas the model experiment has been simulated adopting a calculation time step, *DT*, equal to 0.005 h. For a direct comparison with field measurements, the recording time step of the output files has been set to 1 h. The median grain size, $D_{50}$, has been set to 0.3 mm and the closure depth, *h\**, 0.8 m. Hourly wave conditions recorded at the ADCP located 11 km offshore the beach have been adopted as input to the GSb model (Table 2).

**Table 2.** Observed wave conditions from 27 May 2015 to 3 June 2015; M = month, D = day, H = hour, $H_s$ = significant wave height, $T_p$ = peak wave period, $\theta$ = wave direction.

| M | D | H | $H_s$ (m) | $T_p$ (s) | $\theta$ (°) | M | D | H | $H_s$ (m) | $T_p$ (s) | $\theta$ (°) | M | D | H | $H_s$ (m) | $T_p$ (s) | $\theta$ (°) |
|---|---|---|---|---|---|---|---|---|---|---|---|---|---|---|---|---|---|
| 5 | 27 | 7 | 0.54 | 3.7 | 41 | 5 | 29 | 17 | 0.47 | 2.5 | 71 | 6 | 1 | 3 | 0.57 | 3.3 | 36 |
| 5 | 27 | 8 | 0.51 | 3.6 | 51 | 5 | 29 | 18 | 0.49 | 2.9 | 67 | 6 | 1 | 4 | 0.53 | 3.4 | 37 |
| 5 | 27 | 9 | 0.5 | 3.8 | 26 | 5 | 29 | 19 | 0.55 | 3 | 30 | 6 | 1 | 5 | 0.5 | 3.1 | 56 |
| 5 | 27 | 10 | 0.48 | 3.6 | 38 | 5 | 29 | 20 | 0.61 | 3.3 | 45 | 6 | 1 | 6 | 0.4 | 4.5 | 29 |
| 5 | 27 | 11 | 0.5 | 3.4 | 36 | 5 | 29 | 21 | 0.66 | 3.7 | 16 | 6 | 1 | 7 | 0.4 | 4.1 | 34 |
| 5 | 27 | 12 | 0.47 | 3.2 | 26 | 5 | 29 | 22 | 0.71 | 3.4 | 41 | 6 | 1 | 8 | 0.33 | 4.3 | 36 |
| 5 | 27 | 13 | 0.47 | 2.7 | 77 | 5 | 29 | 23 | 0.7 | 3.6 | 65 | 6 | 1 | 9 | 0.32 | 4.3 | 13 |
| 5 | 27 | 14 | 0.42 | 3 | 51 | 5 | 30 | 0 | 0.6 | 3.3 | 62 | 6 | 1 | 10 | 0.28 | 4.3 | 12 |
| 5 | 27 | 15 | 0.38 | 3.8 | 19 | 5 | 30 | 1 | 0.56 | 3.5 | 36 | 6 | 1 | 11 | 0.27 | 3.7 | 358 |
| 5 | 27 | 16 | 0.39 | 4.1 | 43 | 5 | 30 | 2 | 0.59 | 3.9 | 43 | 6 | 1 | 12 | 0.28 | 3.8 | 62 |
| 5 | 27 | 17 | 0.46 | 2.7 | 24 | 5 | 30 | 3 | 0.71 | 3.3 | 52 | 6 | 1 | 13 | 0.31 | 3.5 | 41 |
| 5 | 27 | 18 | 0.62 | 1.4 | 32 | 5 | 30 | 4 | 0.75 | 3.4 | 45 | 6 | 1 | 14 | 0.31 | 3.3 | 48 |
| 5 | 27 | 19 | 0.68 | 3.5 | 28 | 5 | 30 | 5 | 0.76 | 3.5 | 83 | 6 | 1 | 15 | 0.24 | 3.8 | 14 |
| 5 | 27 | 20 | 0.74 | 3.6 | 22 | 5 | 30 | 6 | 0.72 | 3.4 | 8 | 6 | 1 | 16 | 0.24 | 4.1 | 22 |
| 5 | 27 | 21 | 0.8 | 3.7 | 26 | 5 | 30 | 7 | 0.69 | 3.3 | 63 | 6 | 1 | 17 | 0.22 | 4.2 | 358 |
| 5 | 27 | 22 | 0.85 | 3.8 | 24 | 5 | 30 | 8 | 0.64 | 1.4 | 13 | 6 | 1 | 18 | 0.19 | 4.2 | 6 |
| 5 | 27 | 23 | 1.05 | 4.2 | 32 | 5 | 30 | 9 | 0.6 | 3.6 | 71 | 6 | 1 | 19 | 0.3 | 2.5 | 260 |
| 5 | 28 | 0 | 0.84 | 4.5 | 19 | 5 | 30 | 10 | 0.53 | 4.2 | 27 | 6 | 1 | 20 | 0.2 | 4.1 | 11 |
| 5 | 28 | 1 | 0.83 | 3.8 | 55 | 5 | 30 | 11 | 0.49 | 4.2 | 14 | 6 | 1 | 21 | 0.18 | 4.2 | 10 |
| 5 | 28 | 2 | 0.9 | 3.4 | 70 | 5 | 30 | 12 | 0.52 | 2.9 | 45 | 6 | 1 | 22 | 0.26 | 2.3 | 64 |
| 5 | 28 | 3 | 0.85 | 3.3 | 30 | 5 | 30 | 13 | 0.45 | 3.7 | 33 | 6 | 1 | 23 | 0.42 | 2.7 | 22 |
| 5 | 28 | 4 | 0.78 | 3.8 | 32 | 5 | 30 | 14 | 0.46 | 3.8 | 15 | 6 | 2 | 0 | 0.46 | 3.2 | 9 |
| 5 | 28 | 5 | 0.74 | 3.9 | 47 | 5 | 30 | 15 | 0.45 | 1.6 | 53 | 6 | 2 | 1 | 0.41 | 3.2 | 23 |
| 5 | 28 | 6 | 0.66 | 4.3 | 28 | 5 | 30 | 16 | 0.41 | 3.2 | 9 | 6 | 2 | 2 | 0.47 | 5.2 | 357 |
| 5 | 28 | 7 | 0.57 | 1.4 | 32 | 5 | 30 | 17 | 0.44 | 4.3 | 19 | 6 | 2 | 3 | 0.41 | 3.3 | 352 |
| 5 | 28 | 8 | 0.48 | 4.6 | 20 | 5 | 30 | 18 | 0.46 | 2.7 | 17 | 6 | 2 | 4 | 0.35 | 3 | 37 |
| 5 | 28 | 9 | 0.47 | 3.4 | 46 | 5 | 30 | 19 | 0.52 | 3.2 | 56 | 6 | 2 | 5 | 0.31 | 3.8 | 2 |
| 5 | 28 | 10 | 0.43 | 4.5 | 25 | 5 | 30 | 20 | 0.58 | 3 | 14 | 6 | 2 | 6 | 0.27 | 3.4 | 32 |
| 5 | 28 | 11 | 0.39 | 4.2 | 31 | 5 | 30 | 21 | 0.61 | 3.5 | 44 | 6 | 2 | 7 | 0.39 | 1.4 | 11 |
| 5 | 28 | 12 | 0.36 | 3.8 | 25 | 5 | 30 | 22 | 0.67 | 3.7 | 39 | 6 | 2 | 8 | 0.23 | 2.7 | 22 |
| 5 | 28 | 13 | 0.38 | 3.6 | 12 | 5 | 30 | 23 | 0.75 | 3.8 | 32 | 6 | 2 | 9 | 0.21 | 3.2 | 46 |
| 5 | 28 | 14 | 0.37 | 3.9 | 25 | 5 | 31 | 0 | 0.76 | 3.9 | 16 | 6 | 2 | 10 | 0.16 | 3.3 | 28 |
| 5 | 28 | 15 | 0.38 | 4.1 | 18 | 5 | 31 | 1 | 0.71 | 3.7 | 21 | 6 | 2 | 11 | 0.12 | 3 | 66 |
| 5 | 28 | 16 | 0.45 | 2.9 | 43 | 5 | 31 | 2 | 0.78 | 3.8 | 39 | 6 | 2 | 12 | 0.13 | 2.6 | 146 |
| 5 | 28 | 17 | 0.53 | 3.1 | 55 | 5 | 31 | 3 | 0.72 | 3.5 | 32 | 6 | 2 | 13 | 0.2 | 2 | 47 |

**Table 2.** *Cont.*

| M | D | H | $H_s$ (m) | $T_p$ (s) | $\theta$ (°) | M | D | H | $H_s$ (m) | $T_p$ (s) | $\theta$ (°) | M | D | H | $H_s$ (m) | $T_p$ (s) | $\theta$ (°) |
|---|---|---|---|---|---|---|---|---|---|---|---|---|---|---|---|---|---|
| 5 | 28 | 18 | 0.63 | 3.2 | 84 | 5 | 31 | 4 | 0.67 | 3.7 | 28 | 6 | 2 | 14 | 0.21 | 2.1 | 58 |
| 5 | 28 | 19 | 0.63 | 3.5 | 51 | 5 | 31 | 5 | 0.6 | 3.7 | 48 | 6 | 2 | 15 | 0.19 | 2.5 | 239 |
| 5 | 28 | 20 | 0.63 | 3.4 | 20 | 5 | 31 | 6 | 0.54 | 3.4 | 45 | 6 | 2 | 16 | 0.15 | 2.1 | 76 |
| 5 | 28 | 21 | 0.83 | 3.7 | 37 | 5 | 31 | 7 | 0.53 | 3.9 | 360 | 6 | 2 | 17 | 0.27 | 2.1 | 80 |
| 5 | 28 | 22 | 0.77 | 4.1 | 55 | 5 | 31 | 8 | 0.56 | 2.8 | 70 | 6 | 2 | 18 | 0.29 | 1.9 | 41 |
| 5 | 28 | 23 | 0.83 | 3.4 | 54 | 5 | 31 | 9 | 0.51 | 4.2 | 32 | 6 | 2 | 19 | 0.29 | 2.3 | 52 |
| 5 | 29 | 0 | 0.69 | 3.4 | 53 | 5 | 31 | 10 | 0.49 | 3.6 | 32 | 6 | 2 | 20 | 0.29 | 2.4 | 64 |
| 5 | 29 | 1 | 0.78 | 3.6 | 34 | 5 | 31 | 11 | 0.52 | 1.5 | 61 | 6 | 2 | 21 | 0.34 | 2.4 | 62 |
| 5 | 29 | 2 | 0.7 | 3.9 | 19 | 5 | 31 | 12 | 0.43 | 4.5 | 28 | 6 | 2 | 22 | 0.36 | 2.3 | 54 |
| 5 | 29 | 3 | 0.81 | 3.8 | 56 | 5 | 31 | 13 | 0.41 | 4.1 | 31 | 6 | 2 | 23 | 0.36 | 2.6 | 40 |
| 5 | 29 | 4 | 0.82 | 3.9 | 38 | 5 | 31 | 14 | 0.39 | 3.1 | 51 | 6 | 3 | 0 | 0.32 | 2.7 | 69 |
| 5 | 29 | 5 | 0.77 | 4.1 | 30 | 5 | 31 | 15 | 0.35 | 4.8 | 37 | 6 | 3 | 1 | 0.38 | 2.6 | 55 |
| 5 | 29 | 6 | 0.77 | 4.1 | 54 | 5 | 31 | 16 | 0.31 | 3.9 | 41 | 6 | 3 | 2 | 0.32 | 2.7 | 82 |
| 5 | 29 | 7 | 0.76 | 4.1 | 29 | 5 | 31 | 17 | 0.33 | 1.9 | 352 | 6 | 3 | 3 | 0.27 | 2.7 | 353 |
| 5 | 29 | 8 | 0.6 | 3.9 | 43 | 5 | 31 | 18 | 0.43 | 2.7 | 55 | 6 | 3 | 4 | 0.29 | 2.6 | 51 |
| 5 | 29 | 9 | 0.61 | 3.7 | 38 | 5 | 31 | 19 | 0.54 | 3 | 17 | 6 | 3 | 5 | 0.26 | 2.7 | 62 |
| 5 | 29 | 10 | 0.57 | 3.6 | 45 | 5 | 31 | 20 | 0.58 | 3.4 | 49 | 6 | 3 | 6 | 0.27 | 2 | 52 |
| 5 | 29 | 11 | 0.54 | 3.7 | 17 | 5 | 31 | 21 | 0.57 | 3.4 | 53 | 6 | 3 | 7 | 0.26 | 9.4 | 282 |
| 5 | 29 | 12 | 0.52 | 3.2 | 28 | 5 | 31 | 22 | 0.66 | 3.4 | 26 | 6 | 3 | 8 | 0.24 | 2.9 | 53 |
| 5 | 29 | 13 | 0.53 | 3.5 | 57 | 5 | 31 | 23 | 0.65 | 3.5 | 25 | 6 | 3 | 9 | 0.27 | 1.8 | 21 |
| 5 | 29 | 14 | 0.45 | 3.6 | 30 | 6 | 1 | 0 | 0.7 | 3.8 | 25 | 6 | 3 | 10 | 0.24 | 3.4 | 222 |
| 5 | 29 | 15 | 0.39 | 3.3 | 49 | 6 | 1 | 1 | 0.69 | 3.7 | 53 | | | | | | |
| 5 | 29 | 16 | 0.44 | 1.5 | 25 | 6 | 1 | 2 | 0.67 | 3.7 | 49 | | | | | | |

Table 3 shows the values of the *RMSE* (Root Mean Square Error) from the comparison of the observed and calculated (for different $K_{GSb}$ values) shoreline positions, $y_{i,obs}$ and $y_{i,GSb}$, respectively, at 23 h after the groin deployment; the *RMSE* is defined as,

$$RMSE = \sqrt{\frac{\sum_{i=1}^{N}\left(y_{i,GSb} - y_{i,obs}\right)^2}{N}} \qquad (3)$$

where $N$ is the number of transects along the considered shoreline. A $K_{GSb}$ = 0.01 value determined the lowest *RMSE*, showing a better agreement between the calculated and the observed shoreline positions.

**Table 3.** Calculated and observed shoreline positions at 14 transects along the shoreline and relative values of *RMSE* (Root Mean Square Error).

| | Longshore Distance from Groin (m) | −20 | −14 | −10 | −8 | −6 | −4 | −2 | 2 | 4 | 6 | 8 | 10 | 14 | 20 | *RMSE* |
|---|---|---|---|---|---|---|---|---|---|---|---|---|---|---|---|---|
| | **Initial Shoreline** | 11.3 | 11.6 | 11.5 | 11.8 | 11.9 | 12.0 | 12.8 | 12.7 | 13.1 | 13.5 | 13.6 | 13.5 | 14.2 | 14.9 | |
| **Cross-shore Distance from Baseline (m)** | **Field data at 23 h** | 9.9 | 9.3 | 9.1 | 9.1 | 9.2 | 9.3 | 9.5 | 18.1 | 18.5 | 18.3 | 17.8 | 17.5 | 17.1 | 16.9 | |
| | **GSb, $K_{GSb}$ = 0.005** | 10.1 | 9.8 | 9.5 | 9.4 | 9.5 | 10.0 | 11.2 | 15.7 | 16.2 | 16.4 | 16.5 | 16.4 | 16.4 | 16.6 | 3.825 |
| | **GSb, $K_{GSb}$ = 0.01** | 10.1 | 9.8 | 9.4 | 9.4 | 9.4 | 10.2 | 11.4 | 15.8 | 16.3 | 16.4 | 16.4 | 16.4 | 16.4 | 16.6 | 3.817 |
| | **GSb, $K_{GSb}$ = 0.05** | 10.1 | 9.4 | 9.3 | 9.8 | 9.8 | 11.6 | 12.5 | 17.0 | 16.7 | 16.5 | 16.4 | 16.4 | 16.4 | 16.6 | 4.127 |
| | **GSb, $K_{GSb}$ = 0.1** | 10.1 | 8.9 | 10.2 | 11.2 | 11.2 | 12.8 | 13.3 | 18.1 | 16.8 | 16.8 | 16.6 | 16.5 | 16.5 | 16.6 | 5.269 |

Numerical simulations were conducted for two different lateral boundary conditions: pinned or moving lateral boundary. Figure 11 shows the comparison of the simulated shoreline positions obtained after 23 h for the two different lateral boundaries; as expected, the shoreline shapes differ in vicinity of the lateral boundaries but overlap in vicinity of the groin. If a moving lateral boundary condition is selected, the boundary will move a specified distance over a certain time period. GSb lateral boundaries have been selected as moving boundaries.

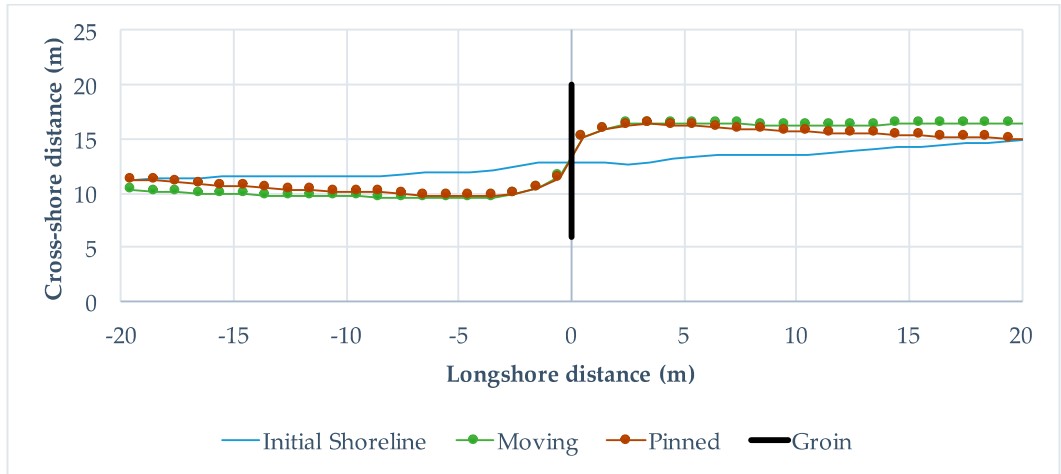

**Figure 11.** Comparison of the simulated shoreline positions obtained after 23 h for the two different lateral boundaries.

The first beach survey was assumed as the initial shoreline position in the numerical model. In particular, at the down-drift ($x$ = 0 m) and up-drift ($x$ = 41 m) boundaries of the computational domain, the observed specific distances from the first beach survey, equal to −1.4 m and 1.8 m, respectively, over a period of 24 h, have been assumed. On the other hand, for the post structure removal condition the computation duration has been extended to 168 h taking as initial condition the survey at $t$ = 24 h.

Figure 12c–g show that the calibrated numerical model satisfactorily predicts the downdrift shoreline evolution, whereas the model is not capable of fully reproducing the shoreline advance, at *t* = 21 h, in the updrift side (Figure 12h).

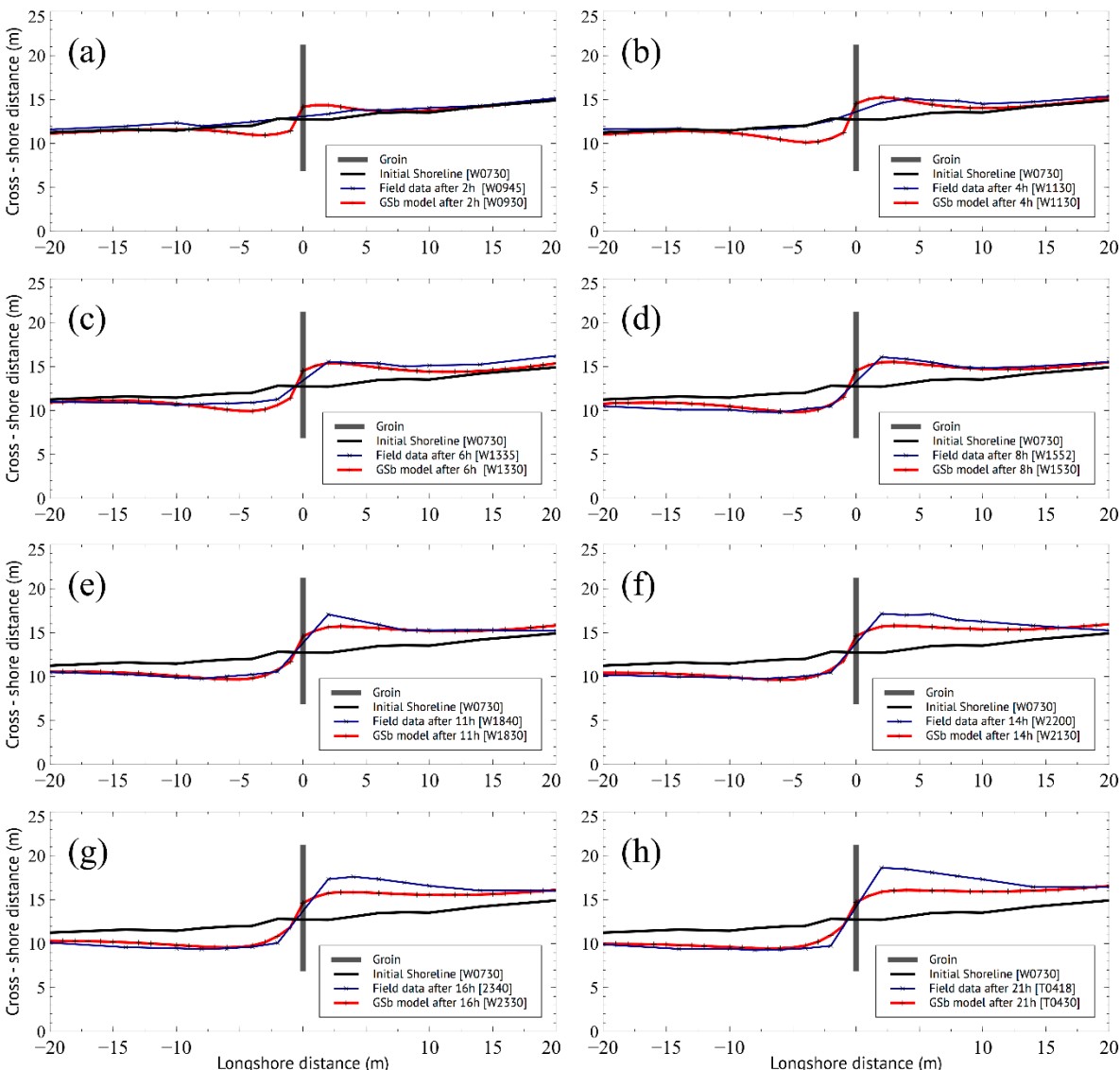

**Figure 12.** Measured and calculated shoreline position (**a**) 2 h, (**b**) 4 h, (**c**) 6 h, (**d**) 8 h, (**e**) 11 h, (**f**) 14 h, (**g**) 16 h and (**h**) 21 h after the structure deployment. Information on the brackets indicates the corresponding weekday (W = Wednesday, T = Thursday) and local time of field data and GSb simulation.

The model capability to predict the shoreline resilience after the structure removal was investigated (verification). The assumed initial shoreline position corresponds to the beach survey performed immediately before the structure removal and the numerical model is run without the structure using the daily mean conditions as measured for the following seven days. The numerical model calculated the drastic change occurring during the first 24 h after the groin removal (Figure 13a–e). Furthermore, it calculated the beach recovery occurring after approximately 7 days (Figure 13f). Therefore, within the framework of the field data gained in the investigation (sea-breeze conditions), the model can be considered as a reliable tool to conduct a numerical study on beach resistance and resilience for the adopted study area.

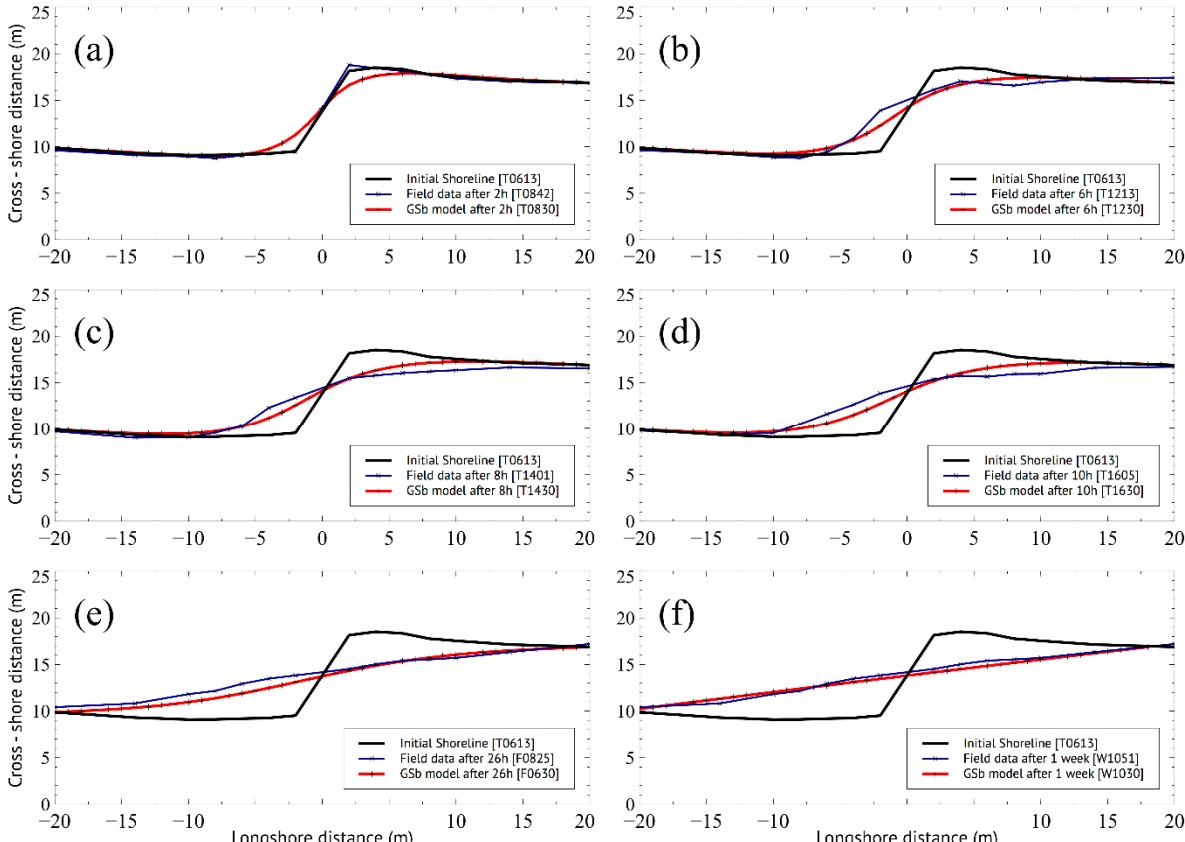

**Figure 13.** Measured and calculated shoreline position (**a**) 2 h, (**b**) 6 h, (**c**) 8 h, (**d**) 10 h, (**e**) 26 h and (**f**) 1 week after the structure removal. Information on the brackets indicates the corresponding weekday (T = Thursday, F = Friday) and local time of field data and GSb simulation.

## 5. Discussion: Shoreline Stability

Once the GSb model has been calibrated and verified, it has been used to investigate the beach resistance and resilience phenomena. The main limitation in the field data is related to the limited alongshore spatial coverage (40 m) of the topographic measurements. Therefore, for the numerical setup, we considered a 10 m long groin located in the middle of a 200 m long shoreline computational domain. For the parametric study we employed different cases (Table 4) encompassing constant (low-energy) wave conditions during a 720 h period. The latter allows us to assess the resistance and resilience sensitivity to different forcing conditions on a longer time-scale basis. For all cases, the shoreline position at the end of the simulation is shown in Figure 14a. A positive correlation between the shoreline distance change and the sediment transport was observed (Figure 14b) and hence $RS(t_0)$ decreases as the sediment transport rate $Q$ [40,41] increases (Figure 14c). Values of $\Delta S(t_0)$ were estimated for wave conditions in Table 4.

**Table 4.** GSb simulated cases to investigate beach resistance to a 10 m groin, for waves of $T_p = 3.5$ s, after 720 h ($t_0$) of simulation.

| Case | $H_{1/50}$ (m) | $\theta$ (°) | $\Delta S(t_0)$ (m) | $RS(t_0)$ | $Q \times 10^{-5}$ (m³/s) |
|---|---|---|---|---|---|
| Test 1A | 0.232 | 15 | 1.33 | 0.765 | 1.52 |
| Test 2A | 0.232 | 30 | 2.28 | 0.629 | 2.45 |
| Test 3A | 0.232 | 45 | 2.62 | 0.585 | 2.46 |
| Test 4A | 0.310 | 15 | 2.47 | 0.604 | 4.03 |
| Test 5A | 0.310 | 30 | 3.75 | 0.454 | 6.47 |
| Test 6A | 0.310 | 45 | 5.11 | 0.324 | 6.50 |

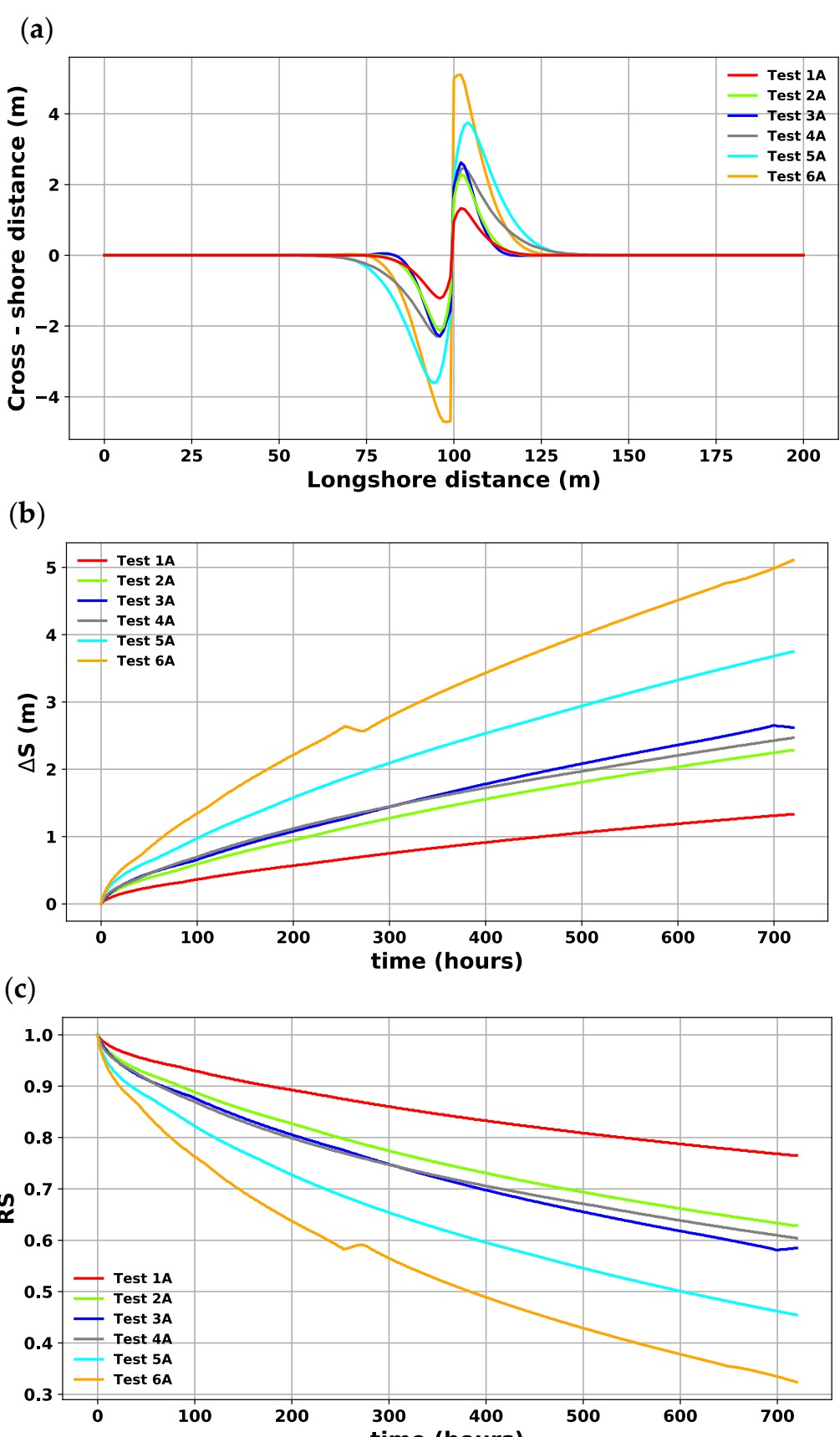

**Figure 14.** (**a**) Shoreline position, owing to the 10 m long groin presence, after 720 h (**b**) the corresponding shoreline distance increases for different wave conditions listed in Table 4 and (**c**) resistance index temporal evolution for different wave conditions listed in Table 4.

The numerical model was also used to simulate the beach recovery phenomena (i.e., resilience) after the structure removal. Thus, the numerical model is initialized with the shoreline from Test 6A at $t_0 = 720$ h (e.g., yellow line in Figure 14a), subjected to different wave forcing conditions (Table 5), without the presence of the structure. The shoreline recovery is significant during the first 24-h for all cases (Figure 15a), consistent with the field observations (Figure 9); afterward, it continues at a lower rate. The numerical results were used to compute the temporal evolution of $RL$ using Equation (2) for each case (Figure 14b). Contrary to beach resistance, numerical results suggest that the beach resilience is not controlled by the longshore transport potential and it depends on the alongshore diffusivity $G$ given by [40,41],

$$G = \frac{2\mu}{(h^* + B)} H_{1/50,b}^{5/2} \cos 2\theta_b \tag{4}$$

where $\mu$ is assumed equal to 0.15 m$^{1/2}$ s$^{-1}$, $h^*$ is the closure depth, $B$ is the berm elevation, $H_{1/50,b}$ is the value of $H_{1/50}$ at breaking [40,41], $\theta_b$ is the wave angle breaking with respect to the mean rectilinear trend of the shoreline. The diffusivity is associated to the longshore spreading of a shoreline perturbation owing to its departure from equilibrium for the existing forcing. The numerical simulations show that, for a given value of $H_{1/50}$, the beach resilience increases as the value of $G$ increases (Figure 15b and Table 5). Therefore, alongshore diffusivity plays an important role on beach resilience in the study area.

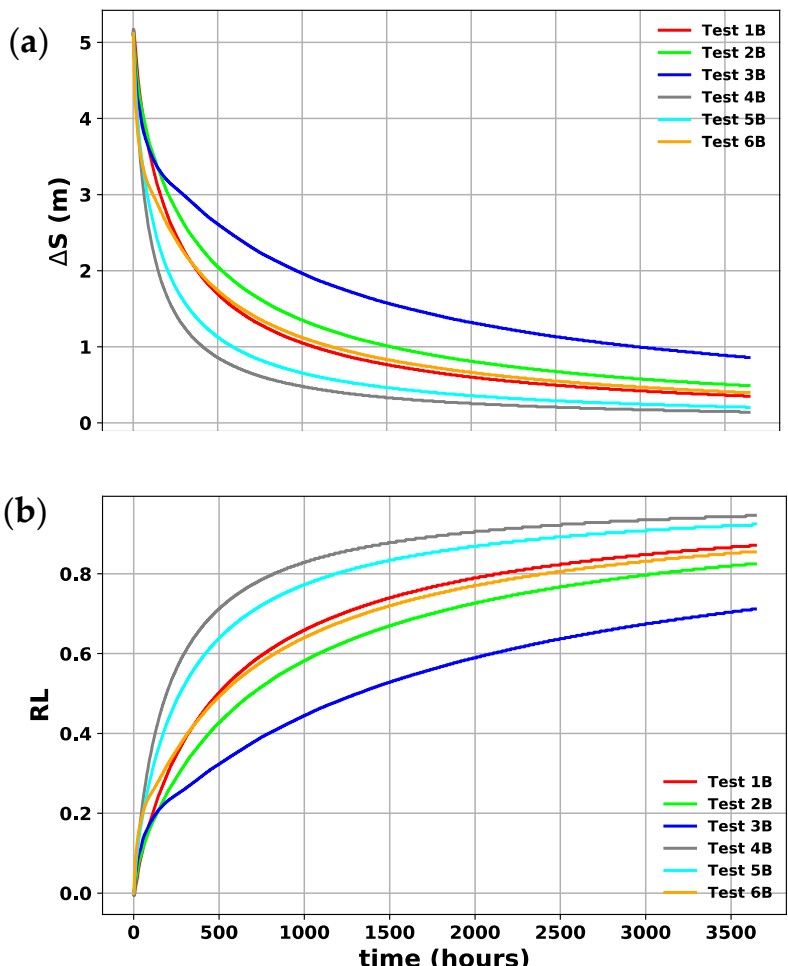

**Figure 15.** Beach resilience after structure removal: (**a**) shoreline change with respect to time after the structure removal and (**b**) resilience index temporal evolution.

**Table 5.** GSb simulated cases to investigate beach resilience associated to the groin removal. The initial condition for all simulations corresponds to the shoreline position for Test 6A at $t$ = 720 h. $\theta_o$ is the deep-water wave angle.

| Case | $H_{1/3}$ (m) | $H_{1/50}$ (m) | $H_{1/50,b}$ (m) | $\theta_o$ (°) | $\theta_b$ (°) | $G \times 10^{-2}$ (m$^2$/s) | $RL(t_i)$ |
|---|---|---|---|---|---|---|---|
| Test 1B | 0.150 | 0.232 | 0.34 | 15 | 3.74 | 0.88 | 0.865 |
| Test 2B | 0.150 | 0.232 | 0.33 | 30 | 6.76 | 0.77 | 0.815 |
| Test 3B | 0.150 | 0.232 | 0.31 | 45 | 8.71 | 0.67 | 0.689 |
| Test 4B | 0.200 | 0.310 | 0.43 | 15 | 3.89 | 1.60 | 0.945 |
| Test 5B | 0.200 | 0.310 | 0.42 | 30 | 7.40 | 1.43 | 0.920 |
| Test 6B | 0.200 | 0.310 | 0.39 | 45 | 10.29 | 1.14 | 0.844 |

## 6. Conclusions

A field and numerical study of shoreline resistance and resilience was conducted on a sea-breeze dominated sandy beach. The following conclusions were found:

(1) Analysis of high- spatial and temporal resolution field observations showed a high growth rate of the perturbation due to a groin disturbance and a lower decaying rate after the disturbance removal.

(2) A new shoreline evolution model was calibrated with field observation and was able to reproduce both the perturbation growth and decay observed in the field.

(3) A parametric numerical study suggests that shoreline resistance decreases with wave conditions enhancing alongshore sediment transport; whereas the resilience increases as a function of the alongshore diffusivity.

**Supplementary Materials:** A demo version of the GSb numerical model, for Mac and Windows systems, has been made available for the scientific community and can be downloaded at: www.scacr.eu and field measurements are available at http://ocse.mx/en/experimento/beach-resilience-to-coastal-structures-brics upon request. A time-lapse video of the beach perturbation due to the temporary groin disturbance is included as supplementary material.

**Author Contributions:** Conceptualization, G.M., A.T.-F., P.A.T.; Writing-Original Draft Preparation, G.M., A.T.-F., G.R.T., A.F.; Writing-Review & Editing, G.M., A.T.-F., G.R.T., A.F., L.L.; Field work, G.M., A.T.-F., J.L., L.P.-A.; Field data Analysis, G.M., Structure design, L.P.-A., J.L., P.A.T., Numerical modeling, G.R.T., A.F., L.L.; Numerical data Analysis, G.R.T., A.F., L.L.; Funding Acquisition, G.M., A.T.-F.

**Funding:** This research was funded by CONACYT through the Cátedras-CONACyT (Project 1146), Investigación Científica Básica (Project 284819) and the Laboratorio Nacional de Resiliencia Costera (Project LN 293354). Additional financial support was provided by PAPIIT DGAPA UNAM (IN101218) and Instituto de Ingeniería UNAM.

**Acknowledgments:** We acknowledge the field support provided by students and researchers at the Laboratorio de Ingeniería y Procesos Costeros at UNAM, especially from Gonzalo Uriel Martín Ruiz, David A. Gracia, Elena Ojeda, Tonatiuh Mendoza, José Carlos Pintado-Patiño, José Alberto Zamora, Daniel Toxtega, Martín Ezquivelzeta, Miguel Ángel Valencia, Paola Espadas, Luis Ángel Gallegos, Daniel Pastrana, Jesús Aragón, Pedro Cabañas, Alejandro Astorga, Enna López, Rafael Meza, Wilmer Rey and Marcos García. Special thanks to Elena Ojeda and David Gracia for providing the time-lapse video of the experiment.

**Conflicts of Interest:** The authors declare no conflict of interest.

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
