# Peer review of "Field and Numerical Study of Resistance and Resilience on a Sea Breeze Dominated Beach in Yucatan (Mexico)"

_water, doi:10.3390/w10121806_

Round 1

Reviewer 1 Report

This paper present a very nice and simple experiment about perturbation of the coastline by a temporary groin. This kind of experiments are scarse but necessary to obtain information to validate coastline evolution models.

Authors also measure the position of the coastline during the experiment and collect a large amount of oceanographic data to provide robustness to the coastline measurements.

Data was analyzed and used to calibrate a one-line model and to evaluate the resistance and resilience of the beach.

The article is well written for publication. But some minor comments for more useful for the readers

1. Figure 1 is not clear, must be improved. This is the only figure on the paper which readers can use to understand the study area, and is not clear, legends are small and some structures that are mentioned on the paper are not located on the map (or are difficult to find) e.g. line 110 port's jetty and sisal pier; line 150 sisal port

2. Regarding the temporary groin, description (lines 135 - 142) and figure 4 must be improved too. I found difficult to correlate description and dimensions on text and figure. Probably locating dimension on the figure will help. Also, a briefly description of the method followed to deploy the structure can be interesting. Including the time required to place and remove the structure.

3. On line 154 is explained that an ultrasonic sensor located on the tower near Sisal Port was used to measure high-frequency wind data, but on table 1, is indicated that wind data was obtained by a weather station; meanwhile the sea level was obtained by the ultrasonic sensor. Which is correct?

4. Authors employ EOF to analyze the field data; I miss a description of the EOF decomposition before spatial and temporal functions are described.

5. Line 259-260, Repeated.

6. Regarding the numerical modeling, to investigate the beach resistance, authors used constant wave conditions lasting for 720 hours. Why these wave conditions? Why constant? Why that long? A justification is needed.

Author Response

Reply to the Reviewer 1

Please see a point-by-point response in the attached file. 

Reviewer 2 Report

The present work presents the results of a field and numerical study aimed to investigate the resistance and resilience on a beach shoreline. In the field test, a permeable groin was implemented in the beach and removed 23hr after. The shoreline change is measured before and after the removal. Such data demonstrate the valuable dynamic shoreline change and recovery due to coastal structure.

As the numerical model, an improved one-line model was developed and calibrated with field survey data. The paper is quite unique and validation is reliable.

There are some unclear parts are remained. Please add several additional experiments.

1)     Line 90; What is “Cold front”?

2)     Line 136; Where is 0.08m hole located? If possible, please indicate it on the figure.

3)     Line 170; DGPS system is capable to measure sea bed profiles during the swell exists? Please explain briefly explain the accuracy the equipment.

4)     Figure 7; Wind data shows a periodical profile. What is a reason that it become so periodical phenomena?

5)     Table 5; H1/3 is usually employed as a representative wave height. Why is H1/5 is employed as the index wave in the table 5?

Author Response

Reply to the Reviewer 2

Please see a point-by-point response in the attached file. 

Reviewer 3 Report

Please find my suggestions and comments on this study in terms of reviewing process for this manuscript.

Brief summary

The paper investigates the shoreline resistance and resilience on a sea-breeze dominated beach based on observations and numerical models in the northern Yucatan peninsula. The main contribution of the study is that resistance to the coastal structure decreases with increasing intensity of wave conditions.

Broad comments 

I think there is a certain amount of work load in front of authors in order to publish this study as a peer-reviewed paper. Broadly speaking, abstract should be rewritten; Discussion section must be included and conclusions section must be rewritten. Regarding these modifications, please find my specific comments and suggestions at the following part.

Specific comments

Please update the title by including the study area.

Abstract:Abstract should definitely be rewritten. It does not follow an uninterrupted story. The first sentence of the abstract for some reason too complicated to start with a research paper (You can actually start Abstract with the first sentence of the introduction. And second sentence can be like the sentence starting from the Line68). I would recommend simplification of that sentence. It is not clear what this study does; Does this study investigate or only presents some results? And I think there is no object in the sentence. 

Please modify this sentence and reorganize whole abstract. A suggestion for the reorganization: Start with the importance of the study, and flow into what has been done in this study and how and summarize results and put some remarks on the conclusions (What is the most important thing that you have concluded in this study).

Also, what is DGPS?

Keyword:please add the study area into the keywords, and remove oneline model from the keywords.

Line 47 : Remove ‘on the other hand’, instead put; However

Line 52: I am not sure whether it is a problem for the journal, but please start sentence with the author’s name, not only with the number 14.

Line 65: same here!

Line 69: The authors start here with the conclusions of the study. Instead of this, I would recommend placing an explanation on what were the research questions in this study and how this study has been done.

Line 98: Same case like Line 65

Please put some explanation on the data with regard to resolution, time period and resources in detail.

Results section is too short to explain the results. Please expand this section. 

Line 167: please delete the firm name Leica.

Discussion Section is definitely needed. Please discuss your results in terms of model constraints, observation limitations and effect on uncertainties on the results in a Discussion subsection.

Conclusions section should only include the conclusions, otherwise it seems redundant. Please do not summarise the work that you have been doing for this study. Only mention what you were able to conclude.

Hope this will help improve the quality of the paper.

Author Response

Reply to the Reviewer 3

Please see a point-by-point response in the attached file. 

Reviewer 4 Report

A very interesting work

Author Response

Reply to the Reviewer 4

Please see a point-by-point response in the attached file. 

Round 2

Reviewer 3 Report

Dear Authors,

thanks for improving the quality of the manuscript. I am still thinking that conclusion section can be more squeezed. There is a redundancy in that section.

Good luck

Author Response

WATER

Reply to the Reviewer 3

“Field and numerical study of resistance and resilience on a sea breeze dominated beach”

by Gabriela Medellín, Alec Torres-Freyermuth, Giuseppe Roberto Tomasicchio, Antonio Francone, Peter A. Tereszkiewicz, Letizia Lusito, Leonardo Palemón-Arcos and José López

Thanks for improving the quality of the manuscript. I am still thinking that conclusion section can be more squeezed. There is a redundancy in that section.

RESPONSE: The paper has been modified according to the reviewer’s comment. More specifically, the conclusions section has been shortened to remove redundancy and be more concise. The revised conclusions section is the following:

“A field and numerical study of shoreline resistance and resilience was conducted on a sea-breeze dominated sandy beach. The following conclusions were found:

(1)   Analysis of high- spatial and temporal resolution field observations showed a high growth rate of the perturbation due to a groin disturbance and a lower decaying rate after the disturbance removal.

(2)   A new shoreline evolution model was calibrated with field observation and was able to reproduce both the perturbation growth and decay observed in the field.

(3)   A parametric numerical study suggests that shoreline resistance decreases with wave conditions enhancing alongshore sediment transport; whereas the resilience increases as a function of the alongshore diffusivity.”